# Impact of Phosphorous Fertilization on Rape and Common Vetch Intercropped Fodder and Soil Phosphorus Dynamics in North China

**Jiahui Qu [1], Lijun Li [1,\*], Peiyi Zhao [2,\*], Dongyu Han [1], Xinyao Zhao [1], Yanli Zhang [1], Li Han [1] and Ying Wang [1]**

[1]  College of Agronomy, Inner Mongolia Agricultural University, No.275, Xin Jian East Street, Hohhot 010019, China
[2]  Inner Mongolia Academy of Agriculture & Animal Husbandry Sciences, Hohhot 010031, China
\*  Correspondence: lijun-li@imau.edu.cn (L.L.); zhpy1972@163.com (P.Z.);
   Tel.: +86-15848154170 (L.L.); +86-13948531236 (P.Z.)

**Abstract:** This study explores the effect of phosphorus (P) fractions, under P addition or not, based on a common vetch-rape model cropping system in alkaline soil. A two year field experiment was conducted at Tuzuo Banner modern agricultural Park in Inner Mongolia, China. Two phosphorus levels, including P0 (no fertilizer) and P45 (45 kg·ha$^{-1}$ P), were performed in common vetch and rape either grown alone or intercropped. We analyzed the changes of the physicochemical properties and phosphorus fractions in the rhizosphere soil. Intercropping enhanced the common vetch and rape yield by 42.05% and 24.91%, on average, compared with corresponding sole cropping on an equivalent area basis. The average land equivalent ratio (LER) was 1.34. Intercropping had a significant AP concentration, of 65.32% and 33.99% at the P0 level, and 62.83% and 36.19% at the P45 level, respectively, compared to that of the sole common vetch and rape. With the application of P, intercropping improved the Resin-Pi and NaHCO$_3$-Pi fraction (61.17%, 87.03% at the P0 level and 96.50%, 41.85% at the P45 level, compared to monocropped common vetch and rape in 2019). The changes in NaOH-Pi and NaOH-Po (except for NaOH-Pi in 2019) showed no significant difference between cropping systems. Intercropping significantly accumulated concentrations of HCl-P, while depleting Residual-P, in 2020. In conclusion, common vetch/rape with the addition of P polyculture stimulated rhizosphere soil P mobilization and had a yield advantage over sole cropping.

**Keywords:** common vetch; intercropping; phosphorus fractions; P application; rape

## 1. Introduction

Phosphorus (P) is the second most indispensable macronutrient for plant growth [1]. Most P removal from soil depends on the parent soil material. As crops remove soil P from within, there is a reduction in the soil P content [2,3], and very little remains when plants use P in the soil [4]. Therefore, supplementing soil P can provide nutritional support for plant growth [5,6]. Farmers routinely apply P fertilizers to increase plant-available soil P concentrations and thus increase crop yield [7]. However, the phenomenon of excessive and irrational application of phosphate fertilizer is widespread in China [8]. When P is added as fertilizer to the soil, it is easy for insoluble phosphate to be fixed by soil or microorganisms. A long-term nonstandard application of phosphate fertilizer thus results in a large amount of P accumulation and a decrease in P use efficiency [3], which thereafter affects soil fertility. In addition, the increase in P discharged through runoff and leaching in farmland ecosystems causes water eutrophication and constitutes a potential threat to the environment [9].

One effective way to solve this problem is by growing plant species that can scavenge recalcitrant soil P. It has been found that certain legume plants, such as white lupin (*Lupinus albus* L.) and faba bean (*Vicia faba* L.), exude large amounts of carboxylates to mobilize and

acquire poorly available soil P [10,11]. The negative effects of legumes on soil P may be attenuated when grown in a mixture with grass [12]. Different plant species can possibly alleviate competition for P in intercropping systems. Durum wheat (*Triticum turgidum durum* L.) and common bean (*Phaseolus vulgaris* L.) exhibited different behaviors in rhizosphere P dynamics [11]. Maize and maize/faba bean intercropping alone depleted the sparingly labile Po fraction, while faba bean alone depleted the labile and moderately labile Po fractions [13]. Common vetch (*Vicia sativa* L.), a kind of leguminous green manure, can fix atmospheric N in root nodules and activate potential nutrient components in soil [14]. Moreover, rape (*Brassica napus* L.) and legumes have much in common in activating insoluble P in the soil. The change in P fractions in pools is affected by many factors. Rhizosphere pH and organic anions exhibit greater contributions than acid phosphatase activity does in enhancing rhizosphere P availability [15]. However, previous studies on P fraction changes only focused on legume-dominated polyculture and root exudates. Phosphatases were emphasized, and very little is known about how soil physiochemical properties respond to P fractions on legume-brassica intercrops. The potential of legume/brassica polyculture in P bioavailability merits investigation.

The cultivated land in China is approximately $1.35 \times 10^8$ hm$^2$, and the P-deficient land is approximately $6.667 \times 10^7$ hm$^2$ [16]. China faces the challenge of feeding its population and livestock and improving the soil environment by implementing sustainable measures in agriculture. The Inner Mongolia agricultural and pastoral staggered area is an important ecological defense line in northern China and, even, across the whole country. It is the hub of material exchange between agricultural areas and pastoral areas. Its geographical environment is suitable for forage grass planting [17]. Accelerating the development of the grass industry in agricultural and pastoral areas and optimizing the planting structure can not only improve the regional economy but also improve the ecological environment, which is an important dynamic for the efficient and sustainable development of agriculture and animal husbandry [18].

In this study, a two year rapeseed-common vetch intercropping experiment with different P applications was used. Our objectives were to (i) evaluate the various soil P fractions in the rhizosphere of monocropped and intercropped plant species and (ii) find the main soil physicochemical parameters that drive the change in soil P fractions in three crop intercropping systems.

## 2. Material and Methods

### 2.1. Site Description

The study was conducted between 2019 and 2020 at the agricultural experimental station (40°56′ N, 110°48′ E), Inner Mongolia Autonomous Region, a typical agropastoral ecotone of north China. This area is characterized by a semiarid and temperate continental monsoon climate, with rainfall concentrated from July to September. The annual precipitation ranges between 300 and 400 mm. The average annual temperature is 7.2 °C, in 2019−2020 (Figure 1), and the frost-free period is approximately 133 days. The annual average sunshine hours are approximately 2952.1 hours. The test station has irrigation conditions and therefore the water factor is not considered in the experiment. The soil type is classified as chestnut soil. The physicochemical properties of the soil before sowing are shown in Table 1.

**Table 1.** Basic characteristics of the tested soil.

| pH | SOM (g/kg) | TN (g/kg) | AN (mg/kg) | TP (g/kg) | AP (mg/kg) | TK (g/kg) | AK (mg/kg) |
|----|-----------|-----------|------------|-----------|------------|-----------|------------|
| 8.45 | 16.56 | 0.53 | 90.4 | 0.54 | 15.51 | 16.64 | 140.2 |

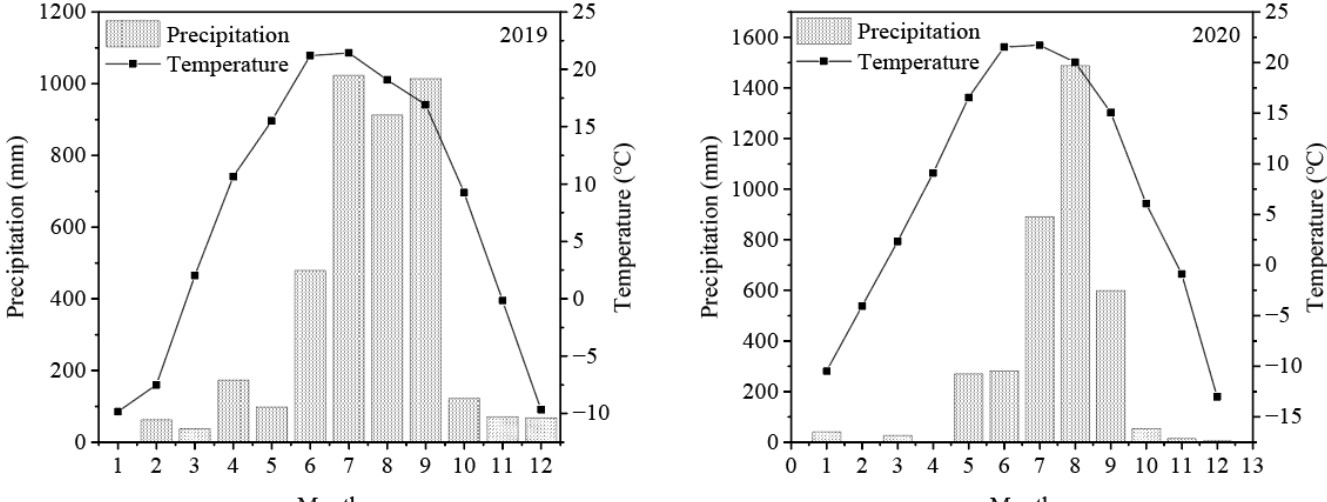

**Figure 1.** Monthly precipitation (bar) and mean air temperature (curve) of the experiment site from 2019 to 2020.

### 2.2. Experimental Design and Crop Management

In 2019, a field experiment was established with rape and common vetch in a completely randomized block design with six blocks. The three cropping patterns were monoculture rape (R), monoculture common vetch (CV), and two rows of rape intercropping with four rows of common vetch (IRCV). All cropping patterns were treated with two P levels; no further P fertilizer was applied (P0), and 45 kg P·ha$^{-1}$ was continuously applied (P45). Each treatment had five replications.

In all of the treatments, the rape and common vetch were cropped in even rows with inter row spacing of 25 cm, and the distance between the rape and adjacent common vetch rows was 25 cm in the intercropping treatments. Each plot had an area of 5 × 6 m$^2$ for monoculture rape and common vetch. Each intercropping plot comprised four strips, and two rows of rape alternating with four rows of common vetch were planted in each strip. Consequently, when planting from the same side of the plot, the last excess row was not sampled, and the intercropping area ratios occupied by rape and common vetch in the IRCV were calculated to be 40%:60% by conversion, based on a unit area. A 0.5 m wide walkway between plots was built to separate the plots from each other.

In all treatments, rape and common vetch were sown at seeding rates of 15 kg·ha$^{-1}$ and 75 kg·ha$^{-1}$, respectively, using 0 kg or 45 kg P·ha$^{-1}$ (as calcium superphosphate), and 120 kg N ha$^{-1}$ (as urea) as basal nutrients. No more fertilizers were applied in the following growth period. All fertilizers were uniformly broadcast and incorporated into the upper 30 cm of the soil before sowing. Weed control was carried out as needed to ensure good germination during the growing season.

### 2.3. Sampling
#### 2.3.1. Forage

To determine the forage sample yield, 1 m plants were hand-harvested at 3 cm above the soil surface, about 86 and 91 days after sowing, in 2019 and 2020. The cuttings were separated from weeds immediately, and the fresh weight was determined. The sample was then dried in an oven (ULM 800, Member GmbH, Schwa Bach, Germany) at 105 °C for 0.5 h and thereafter dried at 80 °C to a constant weight to measure aboveground dry matter biomass.

### 2.3.2. Soil

For soil analyses, we sampled 3 cores (7 cm ⌀, 30 cm) with a soil auger at the first 30 cm of each plot (Figure 2). We removed plant residues from the soil and then sieved at 2 mm. The sample was air-dried to constant weight in a drying room for elemental analyses.

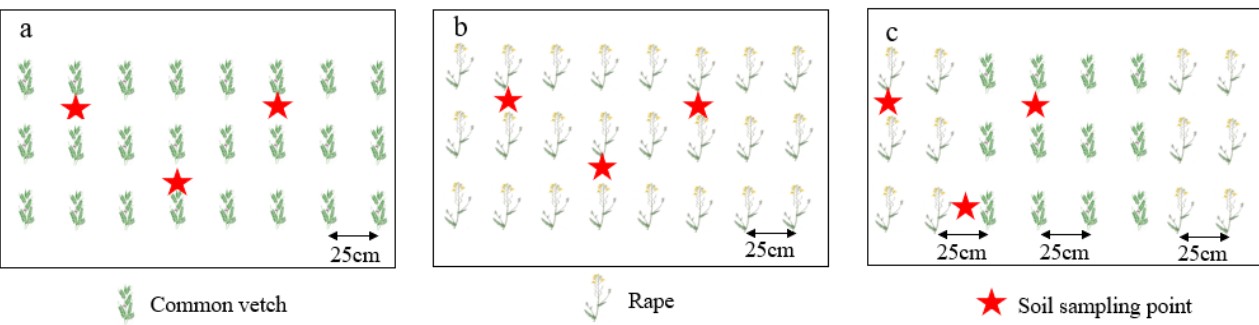

**Figure 2.** Schematic diagram of monoculture common vetch (**a**) monoculture rape (**b**) intercropping of rape and common vetch (**c**) and the location of soil sampling in different treatments.

### 2.4. Soil Analysis

### 2.4.1. General Soil Parameters

The soil pH was measured after the suspension of the soil in water (1:5 weight/volume).

The organic matter (OM) and total nitrogen (TN) contents of the soil samples were determined using an element analyzer (Flash EA, Thermo Electron Corporation, Bremen, Germany). There is no carbonate in the soil. Therefore, the soil C is considered to be exclusively organic. The total phosphorus (TP), organic phosphorus (OP), and inorganic phosphorus (IP) for the bulk soil were determined as a sum of Hedley fractions (see below).

### 2.4.2. Phosphorus Fractionation

The fractionation of soil P was carried out using the Hedley method [19], which adopts the modifications introduced by Tieseen et al. [20]. Next, 0.5 g of dry soil was weighed and extracted sequentially by shaking overnight (16 h) with a solution. (1) First, 30 mL of deionized water and two resin strips (Selemion™ ion exchange membrane; Asahi Glass Co. Ltd., Tokyo, Japan) were used. P was then extracted from these resins by shaking with 20 mL of 0.5 M HCl for 2 h (Resin-P). (2) Secondly, 30 mL of 0.5 M NaHCO$_3$ was used after the pH was adjusted to 8.5 (NaHCO$_3$-P). (3) Subsequently, 30 mL of 0.1 M NaOH (NaOH-P) was used, followed by (4) 20 mL of 1 M HCl (HCl-P). Each step was centrifuged (8000× *g*, 1 min) and filtered. The inorganic P (Pi) concentration in all of the extracts was determined using the Mo-Sb Spectrochrometry method [21]. The total P in each form (Pt) concentration was measured with the same methods after digestion with potassium persulfate and sulfuric acid at hot temperatures. The organic P (Po) in these fractions was calculated by subtracting the Pi from Pt. The residual P, remaining after extraction step (4), was extracted after calcination of the residue for 1 h at 550 °C with 1 M sulfuric acid (H$_2$SO$_4$) for 24 h.

### 2.4.3. Formatting of Mathematical Components

The land equivalent ratio (LER) is often regarded as an indicator of intercropping benefits [22]. The LER was calculated according to:

$$\text{LER} = (\text{Yicv}/\text{Yscv}) + (\text{Yir}/\text{Ysr}) \tag{1}$$

where Yicv and Yir are the forage yields of intercropped common vetch and rape, respectively. Yscv and Ysr are the forage yield of sole common vetch and rape, respectively. If the LER is greater than 1, this indicates an intercropping advantage for yield.

Phosphorus activation coefficient (PAC) is often considered as an indicator of soil phosphorus availability and can be calculated according to the following formula [23]:

$$PAC= (AP (mg/kg)/TP (g/kg) \times 1000) \times 100\% \qquad (2)$$

### 2.5. Statistical Analysis

The statistical analyses were carried out with SPSS (IBM SPSS Statistics 23). Significance was declared at $p < 0.05$. The graphs were constructed using Origin software (Version 8.5; Northampton, MA, USA) and used to draw figures. Redundant analysis was performed with CANOCO (Version 5.0, Ithaca, NY, USA). The partial least squares method (PLS-PM) was used to model the relationship between soil properties and phosphorus fractionation. The path coefficient and the coefficient of determination ($R^2$) in the path model were estimated by R (4.0.3) and verified with the plspm package.

## 3. Results

### 3.1. Effect of Cropping System and P Application Rate on Forage Yields and Land Equivalent Ratios (LERs)

The forage yields for the common vetch sole cropped (CV), common vetch intercropped (ICV), rape sole cropped (R), and rape intercropped (IR) during the two year experiment are presented in Table 2. The average LER was 1.34 (Table 2), indicating that common vetch/rape intercropping had a yield advantage over sole cropping. The data demonstrates that intercropping, with P addition, significantly increased the common vetch and rape yields. Intercropping increased the common vetch yield by 32.40% and 58.01%, compared with that in monocultures at P0 and P45 in 2019 ($p < 0.05$), respectively. The IR at P0 and P45 increased dramatically, by 42.40% and 43.66%, compared with that in the monocultures in 2019 ($p < 0.05$), respectively. P application significantly increased the common vetch yield by 56.31% and 86.55%, the rape yield by 83.30% and 84.93% relative to no P application in 2019 ($p < 0.05$), and by 35.16% and 43.46% and 31.23% and 39.87% in 2020 ($p < 0.05$), respectively. Without P application, the yield of the sole common vetch and rape decreased by 23.82% and 3.80% compared to the corresponding intercrop in 2020, respectively. The intercropped common vetch and rape significantly increased yield with P application by 39.32% and 10.80% in 2020, compared with the corresponding sole crops. Intercropping, on average, enhanced common vetch and rape yield by 42.05% and 24.91%, respectively, compared with the corresponding sole cropping on an equivalent area basis.

**Table 2.** Dry matter yield ($\times 10^3$ kg·hm$^{-2}$) of common vetch and rape and land equivalent ratios (LER) as affected by phosphorus (P)-fertilization rate and cropping system from 2019 to 2020.

| Year | Annual P Rates (kg·hm$^{-2}$) | Common Vetch (kg·hm$^{-2}$) | | Rape (kg·hm$^{-2}$) | | LER |
|---|---|---|---|---|---|---|
| | | Sole Cropped | Intercropped | Sole Cropped | Intercropped | |
| 2019 | 0 | 9.66 ± 1.28 Bd | 12.79 ± 0.21 Bc | 17.43 ± 1.02 Bb | 24.82 ± 0.30 Ba | 1.38 |
| | 45 | 15.10 ± 0.40 Ad | 23.86 ± 0.69 Ac | 31.95 ± 1.48 Ab | 45.90 ± 1.62 Aa | 1.52 |
| 2020 | 0 | 10.01 ± 0.32 Bd | 13.14 ± 0.79 Bc | 22.80 ± 0.63 Bb | 23.70 ± 0.65 Ba | 1.22 |
| | 45 | 13.53 ± 0.94 Ad | 18.85 ± 0.44 Ac | 29.92 ± 0.70 Ab | 33.15 ± 2.40 Aa | 1.23 |
| Mean | | 12.08 | 17.16 | 25.53 | 31.89 | 1.34 |
| Significance of | | *F* | *P* | *F* | *P* | |
| Years (Y) | | 1.06 | 0.31 | 1.00 | 0.323 | |
| P rate (P) | | 173.14 | <0.0001 | 112.46 | <0.0001 | |
| Cropping system (C) | | 108.13 | <0.0001 | 26.81 | <0.0001 | |
| P × C | | 16.06 | <0.0001 | 3.28 | 0.079 | |

Notes: Values are means ± standard errors (n = 5) and yields of intercropped crops are expressed on an equivalent area basis. Values followed by the same lower-case letters between sole crop and intercrop and by the same capital letter between P0 and P45 are not significantly different in a particular year at $p < 0.05$ by LSD. The bold entries indicate *p*-values < 0.05.

### 3.2. Changes in Soil Physiochemical Properties of the Rhizosphere

The soil physiochemical properties were all significantly affected by the P rate and cropping pattern (Table 3). The rhizosphere pH in the monocropped common vetch did not differ from that in the monocropped rape soil. In contrast, intercropping significantly decreased the rhizosphere pH by 0.26 and 0.13 pH units, in 2019, and 0.35 and 0.57 pH units in 2020, respectively, compared with that of the sole rape soil. Compared to the monocultures, the OM content in the intercropped treatments was significantly improved, except for P0 in 2020 (Table 4). With P application, intercropping increased the TN content by 21.05% in 2019 and 15.09% in 2020, compared to that in the sole rape. No difference was observed for the TN at P0 in the different cropping systems in 2019 and 2020 (Table 4). There was a significant interaction between the P rate and the year on TP and AP (Table 3). Compared to the monocropped rape, intercropping had a significant TP concentration of 13.70%, 5.97% at the P0 level in 2019 and 2020 and 5.95% at the P45 level in 2019 (Table 4). Similar to TP, AP showed a trend of higher concentrations under intercropping than under monocultures. Particularly in 2020, intercropping had a significant AP concentration, by 65.32% and 33.99% at the P0 level and 62.83% and 36.19% at the P45 level, respectively, compared to that of sole common vetch and rape (Table 4). The PAC was significantly affected by the P rate, cropping pattern, and their interactions (Table 3). The PAC ranged between 2.16 and 2.73 in 2019 and 2.13–3.81 in 2020 (Table 4).

**Table 3.** Multi-factor analysis of variance of soil physiochemical properties in 2019–2020.

| Factors | pH | OM | TN | TP | AP | PAC |
|---|---|---|---|---|---|---|
| Year (Y) | *** | NS | ** | *** | *** | *** |
| P rate (P) | * | *** | *** | *** | *** | *** |
| Cropping pattern (CP) | *** | *** | *** | *** | *** | *** |
| Y × P | NS | NS | NS | *** | *** | NS |
| Y × CP | NS | NS | NS | * | *** | *** |
| P × CP | NS | NS | ** | *** | NS | NS |
| Y × P × CP | NS | NS | NS | NS | NS | NS |

Notes: NS, *, **, and *** represent $p > 0.05$, $p \leq 0.05$, $p \leq 0.01$, and $p \leq 0.001$, respectively, the same as below.

**Table 4.** Soil physiochemical properties in 2019–2020.

| Year | P | Treatments | pH | OM (g·kg$^{-1}$) | TN (g·kg$^{-1}$) | TP (g·kg$^{-1}$) | AP (mg·kg$^{-1}$) | PAC (%) |
|---|---|---|---|---|---|---|---|---|
| 2019 | P0 | CV | 8.67 ± 0.11 a | 17.06 ± 1.13 b | 0.56 ± 0.02 a | 0.77 ± 0.05 ab | 16.64 ± 1.68 b | 2.16 ± 0.30 b |
| | | R | 8.68 ± 0.07 a | 18.30 ± 1.51 b | 0.54 ± 0.02 a | 0.73 ± 0.04 b | 18.59 ± 1.25 b | 2.56 ± 0.15 a |
| | | IRCV | 8.42 ± 0.15 b | 20.91 ± 2.51 a | 0.54 ± 0.04 a | 0.83 ± 0.02 a | 22.07 ± 2.30 a | 2.67 ± 0.32 a |
| | P45 | CV | 8.52 ± 0.09 ab | 20.09 ± 3.81 b | 0.59 ± 0.02 b | 0.83 ± 0.05 b | 17.85 ± 1.95 b | 2.18 ± 0.35 b |
| | | R | 8.60 ± 0.04 a | 20.22 ± 1.73 b | 0.57 ± 0.05 b | 0.84 ± 0.03 b | 22.52 ± 2.38 a | 2.69 ± 0.35 a |
| | | IRCV | 8.47 ± 0.09 b | 24.08 ± 1.29 a | 0.69 ± 0.04 a | 0.89 ± 0.01 a | 24.35 ± 0.97 a | 2.73 ± 0.09 a |
| 2020 | P0 | CV | 8.44 ± 0.23 ab | 18.57 ± 1.61 a | 0.54 ± 0.05 a | 0.71 ± 0.02 a | 15.14 ± 0.31 c | 2.13 ± 0.09 c |
| | | R | 8.58 ± 0.12 a | 20.50 ± 2.03 a | 0.50 ± 0.06 a | 0.67 ± 0.02 b | 18.68 ± 0.20 b | 2.78 ± 0.11 b |
| | | IRCV | 8.23 ± 0.21 b | 20.77 ± 1.32 a | 0.52 ± 0.09 a | 0.71 ± 0.01 a | 25.03 ± 0.80 a | 3.54 ± 0.12 a |
| | P45 | CV | 8.40 ± 0.24 a | 19.12 ± 1.57 b | 0.60 ± 0.03 a | 0.79 ± 0.02 b | 20.50 ± 0.34 c | 2.59 ± 0.07 c |
| | | R | 8.51 ± 0.24 a | 22.42 ± 2.41 ab | 0.53 ± 0.03 b | 0.86 ± 0.04 a | 24.51 ± 1.40 b | 2.85 ± 0.28 b |
| | | IRCV | 7.94 ± 0.31 b | 24.21 ± 3.01 a | 0.61 ± 0.06 a | 0.88 ± 0.01 a | 33.38 ± 0.64 a | 3.81 ± 0.07 a |

Notes: The data represent the mean ± SD; n = 5. Values followed by different letters within a column indicate significant differences between sole crop and intercrop under the same phosphorus application level at the $p < 0.05$ level.

### 3.3. Changes in Soil P Fractions in the Rhizosphere

The content of P fractions decreased in the order HCl-P > NaOH-P > Residual-P > NaHCO$_3$-P > Resin-P (Figure 3). The soil Resin-Pi was significantly affected by year, P rate, cropping pattern, and the interaction among them (Table 5). The Resin-Pi in the IRCV was reduced slightly, by 8.83% and 6.12%, in 2019, and 16.16% and 12.46% in 2020, respectively, compared to that in the sole common vetch and rape, but intercropping with P application appreciably improved the Resin-Pi content (Figure 4a,b). The NaHCO$_3$-P

was significantly affected by year, P rate, and their interactions (Table 5). The NaHCO$_3$-Pi fraction was significantly (61.17%, 87.03% at the P0 level and 96.50%, 41.85% at the P45 level in 2019) greater in the intercropped, than in the monocropped, common vetch and rape (Figure 4a). Conversely, the NaHCO$_3$-Po fraction decreased by 21.39% and 6.76% and 24.58% and 42.72% in the intercropped and monocropped common vetch and rape, respectively (Figure 4a). The NaHCO$_3$-Pi fraction was dramatically (43.25% and 32.60% at the P45 level in 2020) greater in the intercropped, than in the monocropped, common vetch and rape (Figure 4b). The NaHCO$_3$-Po fraction increased by 28.92% and 20.35% at the P45 level in the intercropped and monocropped common vetch and rape, respectively (Figure 4b). The changes in NaOH-Pi and NaOH-Po (except for NaOH-Pi in 2019) showed no significant difference between cropping systems (Figure 4a,b). Compared to the monocropped common vetch and rape, the 1 M HCl-Pi concentration of the intercrop was significantly increased 28.66% and 8.61% at the P0 level and 9.87% and 23.01% at the P45 level (Figure 4a), and the 1 M HCl-Pi concentration of the intercrop was decreased 4.77% and 0.84% at the P0 level and −10.40% and 3.77% at the P45 level (Figure 4b). Intercropping significantly accumulated concentrations of HCl-Po and Residual-P, while significantly depleting concentrations of HCl-Pi, in 2019 (Figure 4a). Intercropping significantly accumulated concentrations of HCl-P, while significantly depleting Residual-P, in 2020 (Figure 4b).

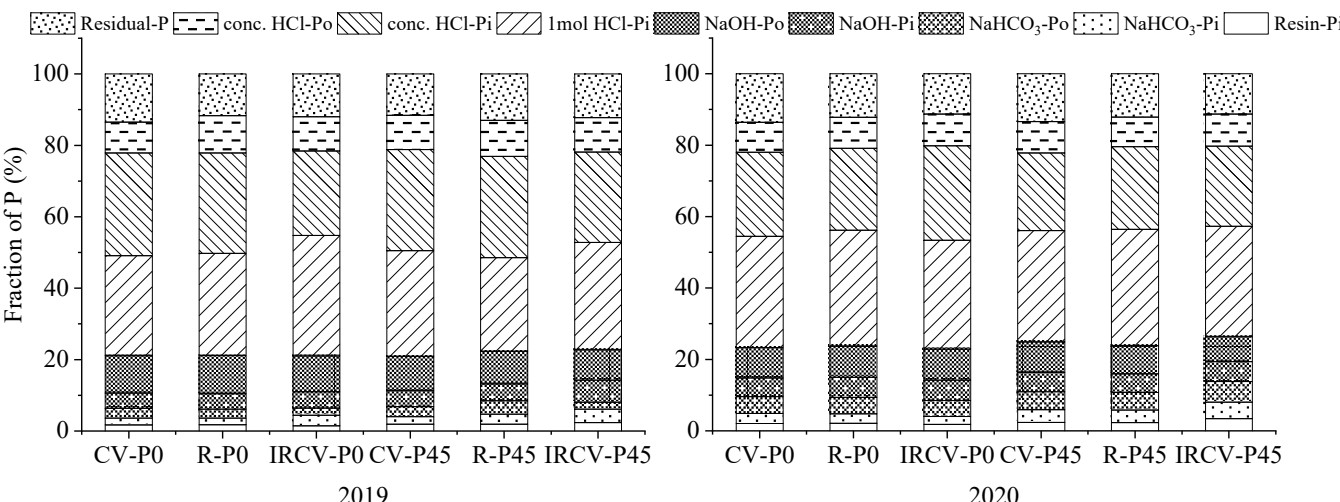

**Figure 3.** Concentration and proportional distribution of soil P fraction in response to different cropping patterns.

**Table 5.** Multi-factor analysis of variance of soil P fractions in 2019–2020.

| Factors | Resin-P | NaHCO$_3$-Pi | NaHCO$_3$-Po | NaOH-Pi | NaOH-Po | 1 M HCl-Pi | conc. HCl-Pi | conc. HCl-Po | Residual-P |
|---|---|---|---|---|---|---|---|---|---|
| Year (Y) | *** | *** | *** | ** | *** | NS | *** | *** | ** |
| P rate (P) | *** | *** | *** | *** | NS | *** | *** | *** | *** |
| Cropping pattern (CP) | *** | NS | *** | *** | NS | *** | NS | *** | ** |
| Y × P | *** | *** | *** | NS | ** | *** | * | NS | * |
| Y × CP | NS | *** | *** | *** | NS | *** | * | *** | *** |
| P × CP | *** | *** | *** | *** | * | *** | *** | NS | *** |
| Y × P × CP | ** | *** | *** | * | NS | *** | *** | NS | NS |

Notes: NS, *, **, and *** represent $p > 0.05$, $p \leq 0.05$, $p \leq 0.01$, and $p \leq 0.001$, respectively.

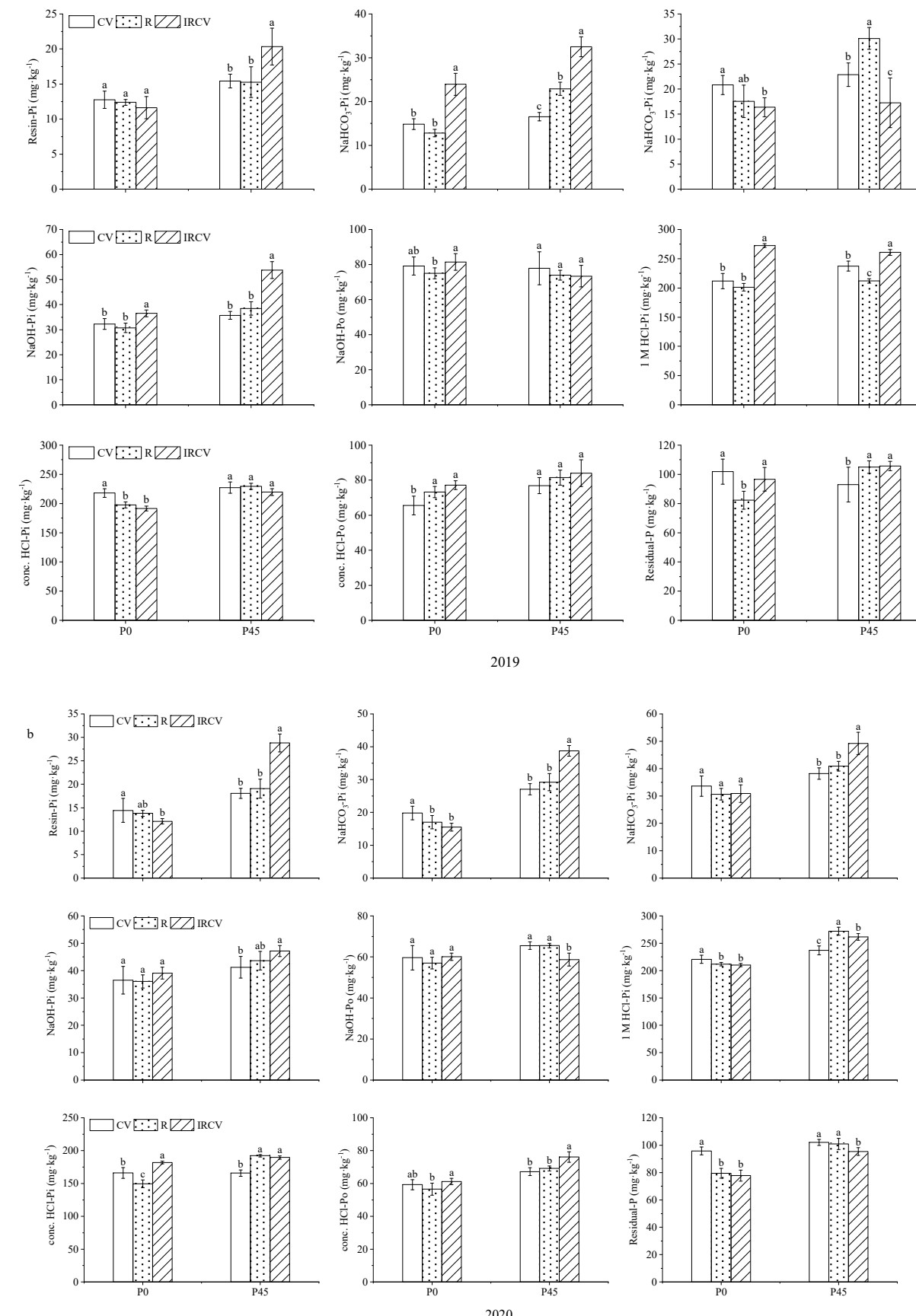

**Figure 4.** Comparisons of P fraction in different cropping patterns at two P levels in 2019 (**a**) and 2020 (**b**).

### 3.4. Relationships between P Fractions and pH, OM, TN, TP, AP, and PAC

Redundancy analyses (RDA) showed that pH, OM, TN, TP, AP, and PAC could explain 78.68% and 9.05% of the variation in P fractions (Figure 5). Among all the con-strained variables, TP ($F$ = 95.9, $P$ = 0.002), AP ($F$ = 5.1, $P$ = 0.002), PAC ($F$ = 18.1, $P$ = 0.002), and pH ($F$ = 3.2, $P$ = 0.012) had a significant impact on the P fractions. The P fractions (except for Residual-P and NaOH-Po) were significantly and negatively associated with pH. The Resin-P, NaHCO$_3$-Pi, NaHCO$_3$-Po, NaOH-Pi, 1 M HCl-Pi, and concentrations of HCl-Po were significantly and positively associated with the OM, TN, TP, AP, and PAC concentrations. HCl-Pi was dramatically and positively associated with TP.

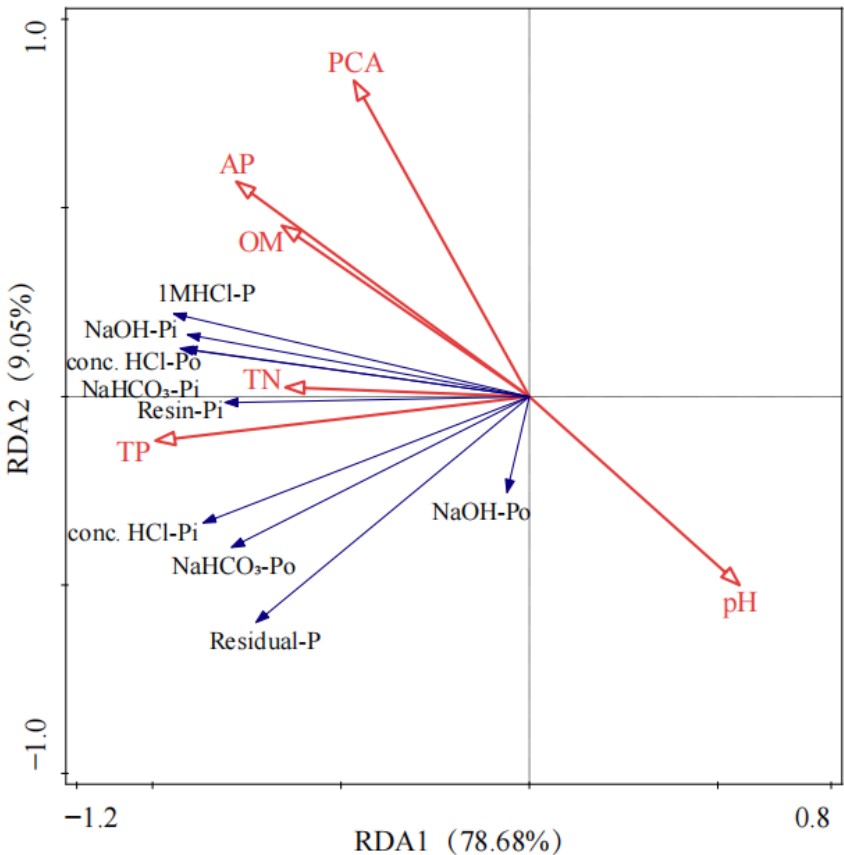

**Figure 5.** Redundancy analysis (RDA) of soil physiochemical properties and P fraction. Note: The data of redundancy analysis were average to that of 2019 and 2020.

### 3.5. PLS-PM Analysis

The PLS-PM explored the relationship between soil pH, soil chemical properties, and P composition (labile P, moderately labile P, nonlabile P and residual P) (Figure 6). This model indicated that the changes in pH had significant and direct negative effects on available P (0.70) and soil labile P (0.20). However, the changes in pH had no direct effect on moderately labile P, nonlabile P and residual P. TP had significant and direct positive effects on labile P, nonlabile P and residual P. We found that TP, AP and OM directly and strongly affected the P composition. However, soil TN had no significant effect on P composition.

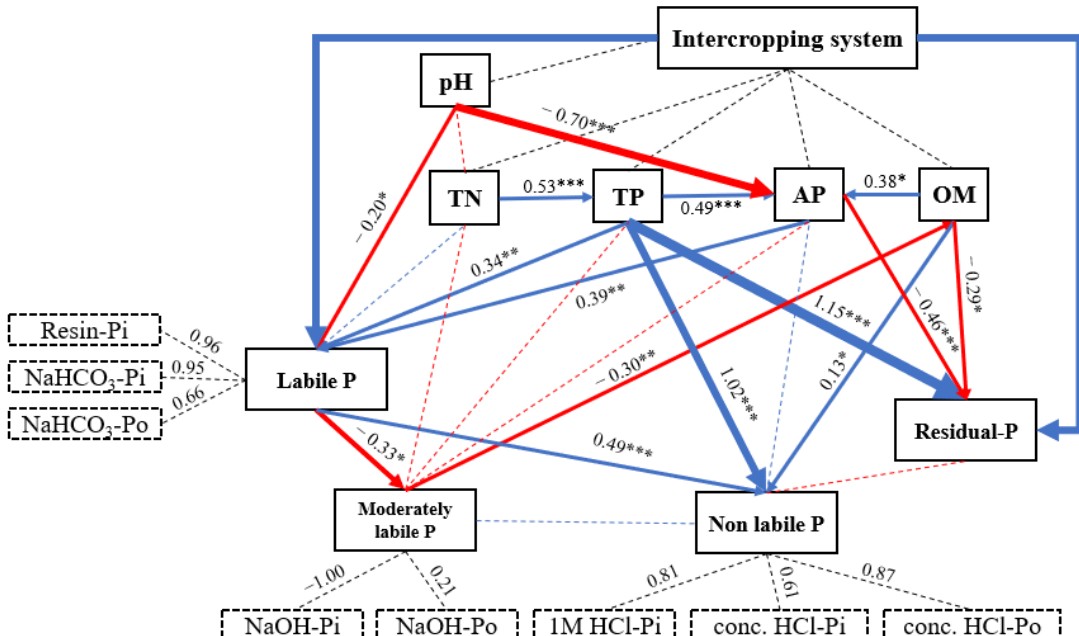

**Figure 6.** Directed graph of the partial least squares path model (PLS-PM). Note: Each box is an observed (i.e., measured) or latent variable (i.e., constructs). Dashed rectangles indicate the load of the different phosphorus forms. Blue arrows indicate positive effects and wider arrows indicate greater effects, and red arrows indicate negative effects. Dashed lines indicate no significant effect ($p > 0.05$). The number of asterisks in the upper right of the number indicates the degree of influence: * $p < 0.05$, ** $p < 0.01$, *** $p < 0.001$. The model is assessed using the Goodness of Fit (GoF) statistic, and the GoF value was 0.59.

## 4. Discussion

In the present study, we found an increase in the aboveground biomass in common vetch/rape intercrops (Table 2). This result contradicts the view that the dry matter of wheat/maize is significantly lower than that of monocropped maize [24]. Previous studies have explained that the process of intercropping system growth and production are essentially driven by niche complementarity and interspecific competition, which is highly related to the species in the community [25]. However, a number of studies have reported consistent results [11,26,27]. For example, Liu found *F. multifora–A. paniculata* intercropping significantly increased the yield compared with sole *F. multifora* by improving the ecological environment and soil quality of rhizosphere soil [28]. The complementary features in morphology and ecological functions between sudangrass and alfalfa prompted productivity [29]. Moreover, additional P is essential for increasing biomass. The beneficial effects of P addition promoted legumes to fix more atmospheric nitrogen and increased the productivity of the inter-crops utilized in photosynthesis, which resulted in maximum forage yield [30,31]. Consequently, common vetch/rape intercropping, when P was provided in a reasonable amount, facilitated the production of desirable forages, which contributes to livestock production.

It has been reported that legumes can generally alleviate soil nitrogen (N) deficiency through symbiotic biological $N_2$ fixation and alleviate P deficiency by changing the soil pH of the root zone [32,33]. In the present study, the TN concentration in the sole common vetch was higher than that in the sole rape. The lack of a significant difference between the sole common vetch and intercropped vetch may be due to competition between common vetch and rape that reduced the N concentration and increased the P in compensation. Intercropping increased the OM concentration, presumably because both common vetch and rape are green manure crops, which are beneficial to soil fertility. After two years of crop growth, the TP and AP concentrations showed a decreasing trend without P application because the cropping systems mainly depleted the P concentration that is available to plants.

Conversely, the P concentration accumulated with the P input. This finding is similar to the dynamics of the phosphorus fraction in common bean/durum wheat intercropping systems [11].

Compared with the monoculture, the common vetch/rape polyculture stimulated rhizo-sphere soil P mobilization in both the common vetch and rape, with significantly increased rhizosphere soil P concentrations (Figure 4). Resin-Pi, NaHCO$_3$-Pi, and NaOH-Pi were significantly increased, particularly Resin-Pi and NaHCO$_3$-Pi, and represented the most available fractions for plant uptake [20,34]. In comparison, the rhizosphere pH decreased by 0.26 and 0.13 pH units in 2019, and 0.35 and 0.57 pH units in 2020, in the intercropping system, respectively. This was significantly less than that in the rhizosphere of the monocropped common vetch (Table 4). In the current experimental design, it was difficult to distinguish between the contributions of common vetch and rape to such pH changes. However, we may assume that the rape made little contribution to these changes, as the monocropped rape did not produce any significant changes in the rhizosphere pH. Intercropping significantly depleted the NaHCO$_3$-Po and 1 M HCl-Pi fractions, especially at the P0 level. This might be caused by the labile P pools in the initial soil that have not yet met the P demand for common vetch and rape growth. The residual P reduction observed in intercropping in 2020 might be supplemented by the transformation of the P from residual P to inorganic P. This finding is in accordance with a study by Li [11].

We used RDA to evaluate the relative importance of rhizosphere pH, OM, TN, TP, AP, and PAC in rhizosphere soil P mobilization. It was found that the P concentration of intercropping was significantly negatively affected by the rhizosphere pH (Figure 5). This finding agrees with most studies [15,35]. On one hand, the reason for this result may be that the organic acids secreted by roots acidify the rhizosphere and improve the utilization of phosphorus; on the other hand, it may also be due to the absorption of cations more than anions in the process of nitrogen fixation in legume crops treated with NH$_4^+$-N fertilizer, which cause the roots to release H$^+$, resulting in rhizosphere acidification [36,37]. In addition, the increase in the Resin-Pi concentration in the intercropping was mainly induced by the TP concentration, and the increase in TP results in the increase in non-labile P and residual-P. (Figure 6). In our study, pH was negatively correlated with labile P and AP; AP, in the treatment at pH 8.40, compared to the treatment at pH 7.94, coincided with a significant increase in labile P (Table 4, Figure 4b). This view disagrees with Bouray because the pH value is below seven [38].

Cultivated pasture is an important part of ecological restoration and sustainable agriculture. The reasonable combination of forage species with complementary traits can benefit livestock and conserve the natural pastures, as well as reduce grazing pressure with higher productivity. Meanwhile, increased forage production could compensate for the insufficiency of available forage. This study indicates that legume/Brassica intercropping offers an effective practice to realize sustainable agricultural development, whether with the addition of P or not. In agricultural soils, the continuous application of P fertilizer not only increases plant-available P forms (labile) and sparingly soluble P forms (mod-labile), but also decreases nonlabile P forms [39,40].

## 5. Conclusions

The present study showed that the common vetch/rape intercropping system increased the yield markedly, with a greater average yield than the corresponding sole crops by 42.05% and 24.91%, respectively, especially with P application. The cropping patterns affected the P fractions over the course of two years of cultivation and significantly depleted the Resin-Pi and NaHCO$_3$-Pi fractions and their concentrations. The HCl-P fraction accumulated without P input. In the case of P fertilization, monocropping or intercropping showed the greatest accumulation of Resin-Pi, NaHCO$_3$-P, NaOH-Pi, and 1 M HCl-Pi fractions. The changes in the soil P fractions that were investigated, with or without P fertilizer application, suggested NaOH-Po, 1 M HCl-Pi and a concentration of HCl-P is a potential source of plant-available P that can be used by common vetch/rape intercropping.

**Author Contributions:** J.Q., L.L. and P.Z. conceived and designed the experiments; D.H. and Y.Z. formed the experiments; X.Z., L.H. and Y.W. performed the statistical analysis; J.Q. and L.L. wrote the paper. All authors have read and agreed to the published version of the manuscript.

**Funding:** This research was funded by the Inner Mongolia science and Technology major special project, Research on rational Utilization technology and Integration mode of water Resources in dryland Area (2020ZD0005-0401).

**Institutional Review Board Statement:** Not applicable.

**Informed Consent Statement:** Not applicable.

**Data Availability Statement:** Not applicable.

**Acknowledgments:** We would like to thank the Farming ecological research Team for field and data collection.

**Conflicts of Interest:** The authors declare no conflict of interest.

**Abbreviations**

| | |
|---|---|
| AP | Available phosphorus |
| LER | Land equivalent ratio |
| OM | Organic matter |
| PAC | Phosphorus activation coefficient |
| TN | Total nitrogen |
| TP | Total phosphorus |

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
