# Peer review of "Impact of Phosphorous Fertilization on Rape and Common Vetch Intercropped Fodder and Soil Phosphorus Dynamics in North China"

_agriculture, doi:10.3390/agriculture12111949_

Round 1

Reviewer 1 Report

The manuscript “Changes in soil phosphorus fractions in response to the phosphorus application and intercropping of rape and common vetch in North China” exploring the effect of phosphorus fractions under phosphorous addition on a common vetch-rape model cropping system in North China.

The manuscript is addressing an interesting aspect of soil phosphorous pool under intercropping system by adding chemical phosphorous in soil. Vicia sativa, known as the common vetch, garden vetch, tare or simply vetch, is a nitrogen-fixing leguminous plant in the family Fabaceae. It is likely native to North Africa, Western Asia and Europe, but is now naturalized in temperate and subtropical area. I feel nitrogen fixation plants in cropping systems would play their role in development of soil fertility gradient. The manuscript is quite interesting for the readers and would contribute to nutrient interactions in soil under field conditions. Please find below some suggestions to improve the manuscript and increase its readability:

            Title:

1. Title of MS “Changes in soil phosphorus fractions in response to the phosphorus application and intercropping of rape and common vetch in North China” is tittle confusing and may be changed as “Impact of phosphorous fertilization on rape and common vetch intercropped fodder and soil phosphorus dynamics in North China”. It’s a suggestion but authors can keep the same title.

Abstract:

2.     Please rewrite the abstract to develop. An abstract is a paragraph that provides readers with a quick overview of your essay or report and its organization The abstract should be a single paragraph and should follow the style of structured abstracts, but without headings. So please remove the headings and convert current abstract into 200 words structured, comprehension paragraph.

3.     Page #1, line # 11-12, “under P addition or not in alkaline soil based on a common vetch-rape model cropping system” is quite confusing sentence especially not in alkaline soil based???? meanings are not clear so rephrase it. Please try to use simple sentences in abstract.

4.     Page #1, line # 16-17, Intercropped common vetch enhanced yield by 42.05% and 24.91% compared with corresponding sole cropping on an equivalent area basis. How this enhanced yield % is calculated? According to table 2, 23-52% yield increase was due to intercropping in both years. Please recheck.

5.     Page #1, line # 16-17, Compared to monoculture, common vetch/rape polyculture stimulated rhizosphere soil P mobilization of both common vetch and rape with significantly increased soil P concentration. How??? According to table 4, 22.52±2.38 was AP in 2019 in rape sole cropping while its value was 24.35±0.97 under intercropping. High standard errors (2.38) in sole cropping can bring its AP 22.52 to 24.90 mgkg-1, not so…….

6.     Page #1, line # 20, add a concluding line on crop performance as well.    

Introduction:

7.     Citation in introduction should be converted into journal requirements.

M&M:

8.     Page #3, line # 111, does 120 N kg ha-1 was applied in no P plots? If not mention it somewhere in M&M

9.     Page #3, line # 119, “To determine the yield, 1 m plants were hand-harvested at 3 cm above the soil surface.” How many days after planting the forage sample were taken? Please indicate it in this section.

1.  Page #4, line # 137, Total P, OP, and IP, please write complete names at first use for easy understanding of nonfamiliar readers.

1.  Page #4, line # 161, Yscv and Ysr are the yield, should be written as forage yield.

1.  Page #5, line # 166, PAC= (AP (mg/kg)/TP (g/kg) × 1000) × 100% (2), please indicate AP, TP, if AP is available P and TP is total P, earlier you mentioned total P as Pt, use same abbreviation in whole MS.

Results:

1.  Page #5, line # 185-195, please recheck the % increase values, I have doubts about some of values.

1.  Page #5, line # 197, Table 2, please indicated the dry matter yield in the table as well, table should be self-explanatory for the readers.

Discussion

1.   Page #10, line # 316 “soled common vetch was higher than that in soled rape” change soled common vetch and soled rape as sole common vetch and sole rape.

1.  Page #10, line # 343-344, P concentration of intercropping was significantly and negatively influenced by the rhizosphere pH, This finding agrees with most studies, i think write the reason why negatively influenced instead of writing agrees the finding of bla bla..

These are my submission. All the best for revisions.

Author Response

Response Letter

Title: Impact of phosphorous fertilization on rape and common vetch intercropped fodder and soil phosphorus dynamics in North China

Journal: Agriculture

Manuscript ID: 2018985

Reviewer #1:

Dear Reviewer:

We really appreciate your efforts and comments on our manuscript, which have significantly improved this manuscript. We have carefully considered the suggestion of Reviewer and make some changes. The red part that has been revised according to your comments. Revision notes, point-to-point, are given as follows:

Comment: Title of MS “Changes in soil phosphorus fractions in response to the phosphorus application and intercropping of rape and common vetch in North China” is tittle confusing and may be changed as “Impact of phosphorous fertilization on rape and common vetch intercropped fodder and soil phosphorus dynamics in North China”. It’s a suggestion but authors can keep the same title.

Response: Thank you for your comments, we agree with your suggestion and have revised the title.

Abstract:

Comment: Please rewrite the abstract to develop. An abstract is a paragraph that provides readers with a quick overview of your essay or report and its organization. The abstract should be a single paragraph and should follow the style of structured abstracts, but without headings. So please remove the headings and convert the current abstract into 200 words structured, comprehension paragraph.

Response: Thank you for your comments. We have rewritten the abstract section. We hope the new description can meet your requirements.

Comment: Page #1, line # 11-12, “under P addition or not in alkaline soil based on a common vetch-rape model cropping system” is quite confusing sentence especially not in alkaline soil based???? meanings are not clear so rephrase it. Please try to use simple sentences in abstract.

Response: Thank you for your comment and we are sorry that we did not express clearly, the original intention of our experiment is to express the intercropping of common vetch / rape in alkaline soil and the monoculture of both, with or without phosphorus application. We have revised in the article and hope the new description is easy to understand.

Comment: Page #1, line # 16-17, Intercropped common vetch enhanced yield by 42.05% and 24.91% compared with corresponding sole cropping on an equivalent area basis. How this enhanced yield % is calculated? According to table 2, 23-52% yield increase was due to intercropping in both years. Please recheck.

Response: Thank you for your comment and we are sorry for the misunderstanding caused by the inaccurate expression. In row 7 of the table, we calculate the average yield, which is compared in the abstract, rather than the specific no-phosphorus or phosphorus treatments in 2019 or 2020. We have corrected the expression in the abstract for a better understanding.

Comment: Page #1, line # 16-17, Compared to monoculture, common vetch/rape polyculture stimulated rhizosphere soil P mobilization of both common vetch and rape with significantly increased soil P concentration. How??? According to table 4, 22.52±2.38 was AP in 2019 in rape sole cropping while its value was 24.35±0.97 under intercropping. High standard errors (2.38) in sole cropping can bring its AP 22.52 to 24.90 mg kg-1, not so…….

Response: Thank you for your comment. We are sorry for the general summary in the abstract, thus ignoring the details. This part has been rewritten and we hope the new description meets your requirements. Revised as instructed line 18-25 in red.

Comment: Page #1, line # 20, add a concluding line on crop performance as well.

Response: Thank you for your suggestion. We made changes on lines 23-25.

Introduction:

Comment: Citation in introduction should be converted into journal requirements.

Response: Thank you for your suggestion. We have revised the citations throughout the article according to the journal requirements.

Comment: Page #3, line # 111, does 120 N kg ha-1 was applied in no P plots? If not mention it somewhere in M&M

Response: Thank you for your comments and 120 N kg ha-1 was applied in every plots, including in no P plots. In line 163, we made a small change in the hope that the new expression would not cause ambiguity

Comment: Page #3, line # 119, “To determine the yield, 1 m plants were hand-harvested at 3 cm above the soil surface.” How many days after planting the forage sample were taken? Please indicate it in this section.

Response: Thank you for your comments. We have indicated in line 171 and 172 in this section.

Comment: Page #4, line # 137, Total P, OP, and IP, please write complete names at first use for easy understanding of nonfamiliar readers.

Response: Thank you for your comment. We agree with you and have revised it in the article.

Comment: Page #4, line # 161, Yscv and Ysr are the yield, should be written as forage yield.

Response: Thank you for your comment. We agree with you and have revised it in the article.

Comment: Page #5, line # 166, PAC= (AP (mg/kg)/TP (g/kg) × 1000) × 100% (2), please indicate AP, TP, if AP is available P and TP is total P, earlier you mentioned total P as Pt, use same abbreviation in whole MS.

Response: Thank you for your comment. We understand what you mean. Thank you for your comments. We understand what you mean. I want to explain the difference between TP and Pt. TP represents the content of total phosphorus in the soil. In lines 203 and 205 marked yellow of the article, Pt represents the total amount of each level of phosphorus speciation. For example, the Pt of NaHCO3 minus NaHCO3-Pi is NaHCO3-Po.

Results:

Comment:Page #5, line # 185-195, please recheck the % increase values, I have doubts about some of values.

Response: Thank you for your comment. We are sorry that we have made a mistake because of carelessness. We have corrected it in the article.

Comment:Page #5, line # 197, Table 2, please indicated the dry matter yield in the table as well, table should be self-explanatory for the readers.

Response: Thank you for your comment and we have indicated the dry matter yield in the table 2.

Discussion:

Comment:Page #10, line # 316 “soled common vetch was higher than that in soled rape” change soled common vetch and soled rape as sole common vetch and sole rape.

Response: Thank you for your comment and we have changed.

Comment:Page #10, line # 343-344, P concentration of intercropping was significantly and negatively influenced by the rhizosphere pH, This finding agrees with most studies, i think write the reason why negatively influenced instead of writing agrees the finding of bla bla..

Response: Thank you for your comment and we have written the reason why rhizosphere pH negatively influenced P concentration. Revised as instructed line 454-459 in red

Reviewer 2 Report

Dear authors,

This is an interesting piece of scientific addition to the particular field of science. All the various sections of the paper are well written and organized.

General comment: it appears that this study entertains only changes in soil P fraction. It gives due emphasis to yields and LERs as well. The same was not either mentioned in the two objectives of the study. Hence, there might be a need for modifying the research topic in a way accomodating the content of the paper.

Specific comments:

I would like to bring to your attention the following points;

1. The conclusion in the abstract (line 20-23) appears does not match the major findings. Nothing is mentioned in the major results about Resin-Pi and NaHCo2-Pi................ In addition, the second objective of the research is not reflected in the abstract.

2. A good number of literature used to support the Introduction part are too old to show the present reality in the particular field

3. The research gap is not clearly is not indicated to show the novelity of the particular study

3. The physical map of the study might be important. If possible, i advise its inclusion.

The source for the information in Table 1 better be indicated

4. In the main conclusion part, focus is again given to yield and LERs. Other wise, it is perfect.

Author Response

Response Letter

Title: Impact of phosphorous fertilization on rape and common vetch intercropped fodder and soil phosphorus dynamics in North China

Journal: Agriculture

Manuscript ID: 2018985

Reviewer #2:

Dear Reviewers:

Thanks very much for taking your time to review this manuscript. I really appreciate all your comments and suggestions. Please find my itemized responses in below and my revisions in the resubmitted files.

This is an interesting piece of scientific addition to the particular field of science. All the various sections of the paper are well written and organized.

General comment: it appears that this study entertains only changes in soil P fraction. It gives due emphasis to yields and LERs as well. The same was not either mentioned in the two objectives of the study. Hence, there might be a need for modifying the research topic in a way accomodating the content of the paper.

Specific comments:

Comment: The conclusion in the abstract (line 20-23) appears does not match the major findings. Nothing is mentioned in the major results about Resin-Pi and NaHCO3-Pi................ In addition, the second objective of the research is not reflected in the abstract.

Response: Thank you very much for your comments. We fully agree with you and have made a comprehensive revision in abstract.

Comment: A good number of literature used to support the Introduction part are too old to show the present reality in the particular field

Response: Thanks for your suggestion and we have made the modification.

Comment: The research gap is not clearly is not indicated to show the novelity of the particular study. The physical map of the study might be important. If possible, i advise its inclusion. The source for the information in Table 1 better be indicated

Response: Thank you very much for your comments and we understand what you mean. Since the experiment was conducted at one site, no map was provided, and the soil nutrients in Table 1 were personally determined in the course of the experiment. In addition, the gap in research and the novelty of specific research need to be further considered in order to achieve it in future research.

Comment: In the main conclusion part, focus is again given to yield and LERs. Other wise, it is perfect.

Response: Thank you very much for your comments. We've made some changes.

Round 2

Reviewer 1 Report

The comments were addressed quite well. Now the MS is improved and i am satisfied.